# Extra Virgin Olive Oil and Metabolic Diseases

**DOI:** 10.3390/ijms25158117

**Published:** 2024-07-25

**Authors:** Vasilis Tsimihodimos, Ourania Psoma

**Affiliations:** Department of Internal Medicine, School of Medicine, University of Ioannina, 45110 Ioannina, Greece; raniapsoma@gmail.com

**Keywords:** extra virgin olive oil, metabolic diseases, Mediterranean diet, diabetes mellitus, lipid disorders, blood pressure, obesity, cardiovascular disease

## Abstract

Over the last few decades, metabolic syndrome coexisting with cardiovascular disease has evolved into a pandemic, making the need for more food-oriented therapeutic approaches and a redefinition of lifestyle imperative, with the Mediterranean diet being the linchpin of this effort. Extra virgin olive oil (EVOO), the key pillar of the Mediterranean diet and one of the most notorious edible oils worldwide, owes its popularity not only to its characteristic aromas and taste but mainly to a series of beneficial health attributes including anti-diabetic, hypolipidemic, anti-hypertensive and anti-obesity actions. In this narrative review, we aimed to illustrate and enlighten EVOO’s metabolic properties through a pathogenetic approach, investigating its potential role in metabolic and cardiovascular health.

## 1. Introduction

Beyond being the cradle of civilization, the Mediterranean basin brought to light the value of the Mediterranean diet (MedDiet). The MedDiet, one of the most popular and studied dietary patterns worldwide, is mostly oriented towards the consumption of fruits, vegetables, nuts and whole grain cereals, with moderate consumption of fish and poultry and meager intakes of meat, sweets and eggs [1]. The key nutritional component of the MedDiet, as well as the main source of fat, is olive oil, and specifically extra virgin olive oil (EVOO) [2]. Olive oil (OO) is a liquid fat and is essentially the juice of Olea europaea. While other dietary oils demand chemical processing and extraction, OO is acquired by physical processes, by pressing the olive fruit, preserving its valuable ingredients [3]. OO can be characterized as EVOO only if it meets strict and specific criteria as established by the International Olive Council (IOC) [4].

EVOO’s organoleptic characteristics and biological properties can be ascribed to an amalgamation of different components (Table 1), with monounsaturated fatty acids (MUFAs) and especially oleic acid (OA) being the main ingredient [5,6]. Other minor but notable components like phytosterols, tocopherols, squalene and phenolics constitute a small part of the overall composition of EVOO, but they also enhance its flavor and taste as well as its health benefits [7]. The phenolic fraction of EVOO is characterized by great heterogeneity with no less than 36 different phenolic compounds being part of its synthesis [8]. The heterogeneity in its chemical composition is an outcome of different cultivars, various extraction, production and storage techniques and diverse environmental conditions which constitute elementary factors for its quality [9,10,11,12]. This peculiar composition and the anti-inflammatory and antioxidant activities of its components have turned EVOO into a key nutritional factor against neurodegenerating diseases, malignancies, metabolic syndrome and chronic diseases [9].

In this narrative review, we summarize the plethora of findings and the rising evidence regarding EVOO’s valuable properties and endeavor to elucidate their association with metabolic disorders.

## 2. EVOO and Relevant Pathogenic Pathways

EVOO’s nutritional value seems to rely on a conjugation of different pathogenic pathways, including inflammation, oxidative stress, endothelial function, coagulation, as well as influence on gut microbiota (Figure 1).

There is abundant evidence supporting the concept that EVOO, enriched as it is in polyphenols and other minority nutrients, exerts a considerable anti-inflammatory effect. In particular, the ingestion of olive oil phenolic compounds has been associated with a decrease in levels of interleukin-6 (IL-6) and C-reactive protein (CRP) in patients with stable coronary heart disease and it is proposed as a supplementary intervention to the pharmacological agenda [14]. Systemic inflammation and increased levels of CRP and IL-6 can be correlated to increased levels of saturated fatty acids of cell membranes. It is widely known that the ratio between the chains of saturated fatty acids and monosaturated fatty acids of phospholipids can directly affect the physical properties of cell membranes. Specifically, an augmentation in saturated fatty acids composition leads to reduced membrane fluidity, as the phospholipids are able to pack tightly together [15]. In their study, Pacetti et al. proved that EVOO can lower the relative ratio of saturated and monounsaturated species of phosphatidylethanolamine (the metabolism of which is essential for cardiovascular health) in erythrocytes’ membranes, ameliorate the levels of polyunsaturated ones, resulting in a general increase in the level of unsaturated phosphatidylethanolamine and subsequently lower the risk of developing cardiovascular disease [15].

Other characteristic inflammatory markers, Thromboxane B2 (TXB2) and Leukotriene B4 (LTB4), seem to be reduced postprandially due to EVOO’s consumption compared to consumption of olive oil or corn oil [16]. Oleocanthal, one of the compounds that gives EVOO its characteristic intense taste, has an action similar to ibuprofen by inhibiting cyclooxygenases 1 and 2 (COX1-COX2) [17]. In addition, other phenolic compounds (tyrosol and B-sitosterol), hamper the cascade of arachidonic acid (AA) by regulating the release of reactive oxygen species (ROS), indicating a possible role of EVOO in prevention of atherosclerosis [18]. Even ligstroside aglycon, a barely known polyphenol of EVOO, impedes the activation of a nucleotide-binding (NOD)-like receptor (NLRP3) inflammasome and decreases the overexpression of COX2 and microsomal prostaglandin E synthase-1 (mPGEs-1) by interfering and inhibiting the signaling pathways of nuclear factor kappa-light-chain-enhancer of activated B cells (NF-κB), MAP kinases (MAPKs) and janus kinase 2/signal transducer and activator of transcription 3 (JAK2/STAT3) [19]. It is known that MAPKs and NF-κB, activated by tumor necrosis factor-α (TNF-α), reduce insulin signaling, arouse inflammatory responses and interfere with peroxisome proliferator-activated receptor gamma (PPAR-γ) activity, leading to insulin resistance and adipose dysfunction [20]. Scoditti et al. were the first to demonstrate that OA and hydroxytyrosol (HTyr) attenuated TNF-α-mediated suppression of adiponectin secretion, as well as the TNF-α downregulation of PPAR-γ in inflamed human adipocytes [20]. Regarding the inflammatory processes that take place in vascular endothelium, Zoubdane et al. showed that a high phenolic intake is correlated to reduced arterial inflammation and atherosclerotic lesion microcalcification (markers of plaque fragility) in healthy, elderly patients [21].

Except for its anti-inflammatory properties, EVOO has been established as an anti-oxidant based on its chain-breaking, scavenging and chelating actions [22]. The overproduction of ROS, the glycoxidation and other oxidative mechanisms can modify low-density lipoprotein (LDL) into oxidized LDL (oxLDL), a known instigator of atherosclerotic events [23]. Phenols in EVOO can bind to LDL particles, mitigating the extent of LDL oxidation in vivo [24]. A MedDiet enriched with high-quality EVOO reduced the levels of 8-hydroxy-2-deoxyguanosine (known marker of oxidative DNA stress), inhibited lipid peroxidation, decreased the levels of IL-6, TNF-α and myeloperoxidase and increased the levels of adiponectin and IL-10 both in obese and normal-weighted adult subjects [25]. In accordance with these findings, several studies have revealed the post-prandial antioxidant activity of EVOO based on the reduction in lipid peroxides in plasma after meals [26,27,28]. In their study, Carnevale et al. not only detected the decrease in diverse oxidative stress biomarkers, but also demonstrated the beneficial role of EVOO in the maintenance of endothelial function by preserving normal levels of soluble vascular cell adhesion molecule-1(sVCAM-1) and sE-selectin (indicators of endothelial impairment), via downregulation of nicotinamide adenine dinucleotide phosphate (NADPH) oxidase [29]. Additionally, hypercholesterolemic patients who followed a MedDiet boosted by EVOO for 4 weeks, showed an improvement in endothelial function compared to patients who followed a saturated fat-enriched diet [30]. Santiago-Fernandez et al. depicted a different protective role of EVOO in endothelial health and atherosclerosis, by examining the impact of triglyceride-rich lipoproteins (TRLs) (which can incorporate EVOO’s components such as tocopherols or carotenoids) on miRNA expression in endothelial cells [31]. In particular, they compared the effect of TRLs isolated from the blood of subjects after a high-fat meal enriched with EVOO or sunflower oil, proving that EVOO-derived TRLs upregulated a set of miRNAs involved in endothelial cell proliferation and angiogenesis regulation such as miR-126-5p [31]. Last but not least, even more processed products such as chocolate enriched by EVOO can play an essential role in endothelial dysfunction by upregulating endothelial progenitor cells (EPCs), molecules known for their pivotal role in vascular integrity [32].

EVOO seems to play an important part in the coagulation process as well. Regular consumption of it appears to restrain platelet adhesion and coagulation cascade by lowering levels of factor VII [33], factor von Willebrand and plasminogen activator inhibitor-1 (PAI-1) [13].

The emerging study of intestinal microbiota over the last two decades has revealed a possible correlation with metabolic disorders [34], with the role of EVOO in this interplay being also researched. Tenorio et al., in their study, supported the idea that the beneficial health effects of EVOO on metabolic diseases and specifically on arterial hypertension may be associated with analogous alterations of the gut microbiota and especially the possible association with the hormone ghrelin [35]. In rats, a diet supplemented with EVOO reformed gut microbiota profile by increasing β-diversity of their composition and subsequently improved metabolic parameters such as insulin resistance and body weight [36]. Moreover, tyrosol, one of the key components of EVOO, provoked weight loss in mice by modulating gut microbiota and by triggering adipose thermogenesis through increased thermogenic gene expression such as elevated expression of uncoupling protein 1 (UCP1) [37].

## 3. EVOO and Diabetes Mellitus

Diabetes mellitus (DM) is considered a challenging public health problem and one of the most common metabolic disorders, affecting millions of patients throughout the world [38]. The role of EVOO in preventing and confronting DM has been thoroughly investigated. In the Prevención con Dieta Mediterránea (PREDIMED) study, one of the largest dietary intervention trials, participants at high cardiovascular risk were randomly assigned to three groups: those who followed a MedDiet enriched with EVOO, those who followed a MedDiet enriched with nuts and those who consumed a control low-fat diet. The incidence of major cardiovascular events was lower for the first and second group compared to the third one, revealing a possible role of MedDiet in reducing cardiovascular risk [39]. A nested substudy of the PREDIMED trial, exhibited after a median follow-up of 4 years, a 51% reduction in diabetes type 2 (T2D) rates in subjects who followed a MedDiet enriched with EVOO compared to a low-fat diet, suggesting a possible role of EVOO in diabetes prevention [40]. In a different substudy of PREDIMED, the MedDiet enriched with EVOO delayed the addition of new-onset glucose-lowering medications and reduced the rate of insulin initiation in patients with T2D by 12% [41]. Moreover, the substitution of carbohydrates with MUFAs as a main dietary pattern in patients with T2D ameliorated their metabolic profile with a reduction in fasting plasma glucose (FPG) [42,43]. Santangelo et al. were the first to notice that the consumption of polyphenol-rich EVOO on a daily basis is correlated with reduction in FPG as well as glycated hemoglobin (HbA1c), probably due to decrease in visfatin levels, an adipose tissue-derived hormone characterized as a proinflammatory cytokine with a key role in impaired glucose metabolism [44].

Numerous studies support the impact of EVOO’s phenolic compounds on beta-cell health through various pathogenic mechanisms. Oleuropein one of the most abundant polyphenols in EVOO, fostering beta-cell insulin secretion and suppressed cytotoxicity generated by amylin amyloids, the aggregation of which correlates with β-cell dysfunction [45,46]. Another known phenolic compound, tyrosol, hindered endoplasmic reticulum stress-induced apoptosis in beta-cells, by interfering with the Jun N-terminal kinase (JNK) signaling pathway [47]. Furthermore, polyphenolic extracts from Olea Europea impeded cytokine-induced oxidative stress and apoptosis in beta-cells and thus preserved redox homeostasis [48]. Marrano et al. were the first to show that olive oil polyphenols and specifically hydroxytyrosol, tyrosol and apigenin promote beta-cell well-being by stimulating proliferation and insulin biosynthesis and by increasing glucose-stimulated insulin secretion (GSIS) [49]. Beyond their effect on pancreatic beta-cells, the phenolic compounds of EVOO contribute to the inhibition of α-amylase and α-glucosidase and consequently to the control of postprandial hyperglycemia as they delay carbohydrate absorption [50,51,52]. Postprandial hyperglycemia and the accompanying production of ROS are also regulated through a different mechanism. Carnevale et al. exhibited that oleuropein ameliorated postprandial glycemic status by interfering with soluble NADPH oxidase-derived peptide activity (sNox2-dp). Postprandial Nox2 activation leads to elevated levels of ROS which are key regulators of the incretin phenomenon. As a result, oleuropein and subsequently EVOO may act as dipeptidyl-peptidase 4 (DPP-4) inhibitors by hampering DPP-4 production and by enhancing glucagon-like-peptide-1(GLP-1) activity [53]. Bozzetto et al. came to the same conclusions, as they proved that the postprandial glycemic peak after a meal with a high glycemic index in patients with diabetes type 1 (T1D) can be restrained after the addition of EVOO, compared to a similar low-fat meal or a meal supplemented with butter [54], based on EVOO’s capacity to influence gastric emptying and enhance GLP-1 secretion [55]. Bartimoccia et al. indicated that EVOO ameliorates postprandial glucose levels, insulin secretion and GLP-1 levels based on a different pathogenic background. They proved that the addition of EVOO to a MedDiet or chocolate altered gut permeability and consequently metabolic endotoxemia by reducing circulating lipopolysaccharides (LPS) and zonulin (a protein that increases the permeability of tight intestinal junctions), whose levels are inversely associated with levels of GLP-1 [56]. With the exception of T2D and TID patients, EVOO improved postprandial glycemic status for patients with prediabetes [57]. Another study based on an ex vivo human model showed that oleuropein presented an additional antidiabetic action, residing in glucose transporter 2 (GLUT-2) inhibition [58].

The protective role of EVOO is not only limited to the development and control of DM, but also to the prevention of its complications. A post hoc analysis of a cohort of patients with T2D participating in the PREDIMED study revealed that the MedDiet supplemented with EVOO can decrease diabetic retinopathy incidence while the incidence of diabetic nephropathy is reduced insignificantly [59]. On the other hand, a recent randomized controlled trial proved that the EVOO-supplemented MedDiet inhibited the estimated glomerular filtration rate (eGFR) reduction and maintained kidney function compared to a low-fat diet in patients with T2D and coronary heart disease (CHD). Remarkably, patients with mildly impaired eGFR seemed to benefit more from the MedDiet [60]. In an experimental model of T1D performed in rats, 3′,4′-dihydroxyphenylglycol, a phenolic compound of EVOO, showed nephroprotective action with a decrease in urinary protein excretion and glomerular morphological changes, based on its antioxidant properties [61]. In addition to its nephroprotective role, 3′,4′-dihydroxyphenylglycol shielded the retina and brain slices of rats against hypoxia-reoxygeneration in a similar experimental model [62].

Apart from T1D and T2D, EVOO appears to contribute to preventing and managing gestational diabetes mellitus (GDM) too. The St. Carlos GDM prevention study revealed that a prompt dietary intervention in the early stages of pregnancy based on a MedDiet supplemented with EVOO and pistachios can reduce the incidence of GDM, the number of patients who finally require insulin therapy and it can inhibit a plethora of maternal and neonatal complications such as prematurity and emergency caesarean section [63]. A sub-analysis of the above study, exclusively limited to normoglycemic pregnant women, came to confirm the aforementioned results [64], while another prospective universal study proposed the adoption of the MedDiet in the early stages of pregnancy as a first line therapy [65]. The above dietary pattern based on EVOO and nuts seemed to defend women from abnormal glucose regulation and metabolic syndrome (MetS), even 3 years post partum [66]. It is known that in GDM pregnancies, placental levels and expression of PPARs are decreased. An EVOO-enriched diet administrated to women with gestational diabetes led to reduced levels of triglycerides, body weight and pro-inflammatory markers (TNF-α, IL-1β), based on the capacity of MUFAs to act as PPAR ligands which can be transported through the placenta to the fetus, act as PPAR activators and regulate metabolic and anti-inflammatory pathways [67].

## 4. EVOO and Lipid Disorders

The crucial role of the EVOO-enriched MedDiet on the primary prevention of cardiovascular events in high-risk patients as it was represented in the PREDIMED trial [39], can be ascribed to an amelioration of LDL particles size, cytotoxicity and resistance against oxidative stress according to a subsequent sub-analysis [68]. Another PREDIMED report, examining the effect of an EVOO-focused MedDiet on apolipoproteins at three months of intervention, did not manage to prove any change in LDL-cholesterol (LDL-C) levels, but demonstrated notable reductions in plasma ApoB, ApoB/ApoA-I ratio, with an increase in plasma ApoA-I indicating the role of EVOO in ameliorating cardiovascular risk [69].

EVOO’s lipid-lowering effects and eventually its cardiovascular safeguarding role is a combination of distinct biological pathways. One of the main hypolipidemic actions of EVOO is based on its capacity to shield HDL-c from oxidation and to promote cholesterol efflux, which is the first step of reverse cholesterol transport. Through reverse cholesterol transport, cholesterol in peripheral tissues is effluxed into HDL particles and is rerouted back to the liver for excretion, making this procedure one of the key HDL cardioprotective mechanisms. Not only has EVOO protected HDL from oxidative impairment, but it also enhanced ABCA1and ABCG1 protein expression, main factors in cholesterol efflux and HDL genesis [70,71]. This result was enhanced by Otrante et al. who also emphasized that EVOO supports cholesterol efflux through protecting HDL functionality against age-related damage [72]. Moreover, when enhanced with green tea polyphenols, EVOO can increase HDL-cholesterol (HDL-C) levels and reduce the size of atherosclerotic lesions in mice by 20% [73]. In this regard, when compared to a diet based on saturated fats (butter), adherence to an EVOO-enriched diet was correlated with increased levels of HDL-C and decrease in cardiovascular risk in postmenopausal women [74]. On the other hand, in a recent cross-over, randomized controlled trial, EVOO ameliorated HDL-C fraction but not the contribution of HDL to cholesterol efflux [75].

Except for HDL oxidation, EVOO appeared to inhibit LDL oxidation by eliminating the expression of proatherogenic CD40-ligand and its downstream products, compared to olive oil with low phenolic ratio [76]. Perrone et al. exhibited that post-prandial hydroxytyrosol decreased oxidized LDL, triglycerides, malondialdehyde and interfered with the stimulation of catalase, superoxide dismutase 1 and transcription factor 1, genes that are involved in lipid metabolism [77].

Another hypocholesterolemic action of EVOO rests on its phenols’ capacity to impede in vitro the 3-hydroxy-3-methylglutaryl co-enzyme A reductase (HMGCoAR) activity in a dose-related way by increasing its phosphorylation through adenosine monophosphate-activated protein kinase (AMPK) pathways. Furthermore, they managed to augment the LDL receptor protein levels in hepatic human cells and, as a consequence, the uptake of LDL extracellular molecules exerting an hypolipidemic effect [78]. In their study, Ródenas et al. proved that the dietary replacement of an olive oil and sunflower oil blend by EVOO resulted in a decrease in total cholesterol (TC), apo- AII, apo-B, VLDL and all fractions of LDL levels in post-menopausal women, as well as the estimated 10-year cardiovascular risk, implicating again the anti-atherogenic ability of EVOO [79]. For patients undergoing a coronary angiography, consumption of EVOO rich in polyphenols slightly decreased levels of LDL-C and enhanced the LDL-C-induced production of Interleukin 10 (IL-10) [80]. The accumulated evidence of different studies also identified the anti-atherosclerotic and LDL-lowering effects of EVOO in different patient groups and with different combinations of dietary interventions [81,82,83,84].

## 5. EVOO and Blood Pressure

The interplay between EVOO and hypertension is another point worth mentioning. The anti-hypertensive actions of EVOO and specifically of EVOO’s micronutrients seem to be based on nutrigenomic properties. Martín-Peláez et al. in their trial demonstrated that daily ingestion of EVOO rich in phenolic compounds reduced systolic blood pressure (SBP) by inhibiting the expression of genes associated with the renin–angiotensin–aldosterone system such as angiotensin-converting enzyme (ACE) and nuclear receptor subfamily 1 group H member 2 (NR1H2) genes [85]. Loizzo et al., in their paper, proved that phenolic compounds found in EVOO could inhibit ACE’s action [51] and proposed as a possible mechanism the ability of flavonoids to engender chelate complexes with zinc ions within the active center of ACE [86]. Furthermore, another substudy of the milestone interventional PREDIMED trial exhibited that patients who enrolled in both MedDiets, enriched either with EVOO or with nuts, presented lower levels of systolic and diastolic blood pressure (DBP) and higher levels of nitric oxide (NO), a strong vasodilator, indicating another possible anti-hypertensive property of EVOO [87]. Storniolo et al. came to the same conclusions and stated that the blood pressure-lowering effects of EVOO relied on the upregulation of NO and downregulation of caveolin 2 in hypertensive women [88]. Except for increased production of NO, EVOO’s polyphenols appeared also to shield endothelial function and subsequently blood pressure (BP) by attenuating endothelin-1 (ET-1), a known vasoconstrictor peptide [89]. In addition, polyphenol-rich olive oil reduced BP and encountered a series of culprits in endothelial dysfunction such as serum asymmetric dimethylarginine (ADMA), ox-LDL, plasma C-reactive protein, while in parallel, it augmented hyperemic areas after ischemia [90]. On the contrary, in a previous sub-analysis of PREDIMED, the anti-hypertensive effects of a MedDiet supplemented with EVOO or nuts were exerted only in DBP, with levels of SBP showing no difference among the three diets [91]. Another anti-hypertensive mechanism was proposed by D’Agostino et al. who showed that EVOO’s phenols could vasodilate mesenteric arteries in rats by stimulating BKca channels via an augmentation of local intracellular Ca^2+^ level as a consequence of inflow through plasma membrane and release from sarcoplasmic reticulum Ca^2+^ storage [92]. Hidalgo et al. implied in their paper that EVOO’s induced alternations in the gut microbiota of hypertensive rats and especially the increase in specific bacteria was related to a decrease in SBP [93]. Two more randomized clinical trials, emphasized the anti-hypertensive properties of EVOO and its simultaneous metabolic action by reducing weight [94], fasting glucose and total cholesterol respectively [95]. Lastly, Njike et al., in their study concerning patients at risk for T2D, compared the effects of EVOO and refined olive oil without polyphenols on endothelial function and BP and came to different conclusions demonstrating that EVOO’s beneficial action was limited only to endothelial function without a difference in BP levels between the two interventions [96].

## 6. EVOO and Body Weight

As obesity develops into a pandemic, the need to embrace a healthy lifestyle becomes more and more intense, with the MedDiet and specifically EVOO appearing as the nutritional keys to this effort [97]. The EPIC-PANACEA study showed that a high adherence to a Mediterranean diet including EVOO reduced the 5-year risk of becoming overweight or obese [98]. Additionally, a secondary analysis of the PREDIMED trial, showed that after 3 years of intervention, patients who followed a MedDiet supplemented with EVOO obtained higher levels of plasma antioxidant capacity which is correlated with a decrease in body weight [99], while another sub-analysis of the specific study did not demonstrate weight gain; on the contrary, it demonstrated weight loss after increasing dietary energy density based on the MedDiet [100]. Another categorization of obesity has been given by Barrea et al. who dichotomized obese patients into two main categories, metabolically healthy and metabolically unhealthy, and tried to clarify the role of the MedDiet in determining these two phenotypes while using it as a screening tool. They found out that the adoption of a MedDiet, and especially EVOO consumption, protected against the development of a metabolically unhealthy phenotype and could serve as an indicator of patients at high cardiovascular risk [101]. A different category of patients, obese breast cancer survivors, appear to benefit from an olive oil supplemented diet due to weight loss and subsequently lower risk of recurrence [102]. Moreover, Cândido et al. were the first to prove that EVOO’s ingestion reduces body fat due to an energy-restricted program without following a Mediterranean diet pattern [94].

A variety of molecular mechanisms have been implicated with regard to EVOO’s anti-obesity properties, with the phenolic fraction claiming the lion’s share. Olive polyphenols appear to regulate weight gain via a plethora of signaling pathways and biochemical procedures including promotion of lipolysis, inhibition of lipogenesis, suppression of pre-adipocyte differentiation and induction of adiponectin secretion through control of genes expression [103]. Polyphenols can also trigger brown adipose tissue and exert thermogenic actions through different molecular pathways including AMPK, peroxisome proliferator-activated receptor c coactivator-1a (PGC1a) or sirtuin 1 (Sirt1) [104]. Oleuropein augmented UCP-1 expression in brown adipose tissue and reduced levels of visceral fat mass in obese rats functioning as an agonist for transient receptor potential ankyrin subtype 1 (TRPA1) and transient receptor vanilloid potential subtype 1 (TRPV1) which are related to weight control, thermogenesis and hormonal changes [105]. Tyrosol may function as a ligand that interacts with the nuclear hormone receptor peroxisome proliferator-activated receptor alpha (PPAR-α). Downstream genes of PPAR-α, which are associated with the thermogenic activity of fat cells, such as UCP1, iodothyronine deiodinase 2 (DIO2), PGC1a, and PR domain containing 16 (PRDM16), were markedly elevated in both brown adipose tissue and inguinal white adipose tissue of mice following tyrosol administration [37]. Scoditti et al. proved that hydroxytyrosol could reduce chronic inflammation of adipose tissue and subsequently obesity-induced diseases by impeding NF-κB activation, inhibiting the expression of genes stimulated by TNF-a and decreasing the production of ROS [20]. Adipocyte differentiation and proliferation could also be prevented by Tyrosol through converting white adipose tissue to brown adipose tissue and regulating PPARγ-related mechanisms [106].

Except for weight gain, EVOO seems to interact with one of the main obesity-related disorders, the non-alcoholic fatty liver disease (NAFLD). The mechanisms responsible for NAFLD are not fully comprehended yet. The localization of lipopolysaccharides (which constitute the outer membrane of gut microbiota) within the liver cells of patients with NAFLD indicates gut-derived endotoxinemia as one possible cause. In the case of gut dysbiosis, lipopolysaccharides enter the bloodstream and localize within the liver cells, where they interact with Toll-Like Receptor 4 and stimulate liver inflammation. Oleuropein appears to decrease liver inflammation and steatosis by blocking intestinal and liver Toll-like Receptor 4 macrophages and suppressing lipopolysaccharides localization [107]. In patients with NAFLD, a low-calorie diet enriched with EVOO resulted in weight loss and a significant decrease in hepatic enzymes, Alanine Aminotransferase (ALT) and Aspartate Aminotransferase (AST), compared to a diet with typical consumption of olive oil [108]. Furthermore, daily consumption of EVOO with high concentration of oleocanthal for two months in patients with MetS and hepatic steatosis, led to a decrease in body weight, waist circumference and body mass index (BMI) as well as in ALT and fatty liver index, ameliorated abdominal fat distribution and regulated a number of inflammatory cytokines [109]. A more recent study demonstrated that high consumption of EVOO is correlated with a lower prevalence of NAFLD, especially for patients who are already dealing with weight disorders [110].

## 7. Conclusions

Either alone, or as a component of the MedDiet, it has been proven that EVOO presents a multitude of healthy properties and acts as a key parameter of clinical nutrition. This cornerstone of the MedDiet appears to afford protection from a plethora of metabolic disorders including diabetes mellitus, hypertension, obesity and lipid abnormalities and subsequently fortifies cardiovascular health. In this narrative review we present the current data regarding EVOO and its metabolic actions by delineating diverse nutrigenomic studies and by shedding light on signaling pathways and molecular mechanisms.

Perceiving the precise role of EVOO and its components in metabolic health can be an additional useful tool in clinicians’ hands guiding them to more food-based therapeutic decisions. Although there is growing evidence of EVOO’s beneficial properties, considerable fields of research still remain unexplored or unclear. More clinical studies should ai towards pointing out the nutrients of EVOO that interfere with metabolic pathways, leading perhaps to more nutrition-oriented therapeutic approaches and reducing the need for pharmaceutical interventions.

## Figures and Tables

**Figure 1 ijms-25-08117-f001:**
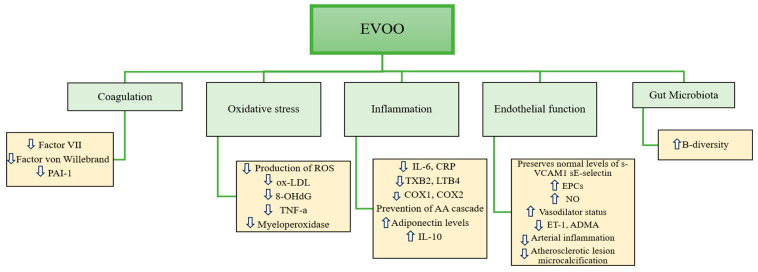
Schematic overview of the numerous biological properties of EVOO.

**Table 1 ijms-25-08117-t001:** Components of EVOO.

Major Components (98–99%)		
Component	Concentration (%)	References
Oleic acid	63–83%	[6,10,11]
Linoleic acid	3.5–21%	[6,10,11]
Palmitic acid	7.5–20%	[6,10,11]
Stearic acid	0.5–5%	[6,10]
Linolenic acid	0–1.5%	[6,10]
**Minor Components (1–2%)**		
**Component**	**Concentration (mg/kg)**	**References**
Sterols	100–250/100 gr	[6,13]
Hydrocarbons (squalene)	200–8260	[5,6]
Polyphenols (1) Sacoiridoids (Oleuropein aglycone, Deacetoxy oleuropein, Oleocanthal and oleacin, Ligstroside aglycone) (2) Phenolics (Hydroxytyrosol, Tyrosol) (3) Phenolic acids (Gallic acid, Ferulic acid, Cinnamic acid, Vanillic acid. Caffeic acid, Syringic acid, Protocatechuic acid, p-Hydroxybenzoic acid, p- and o-coumaric acid (4) Flavonoids (Luteolin, Apigenin) (5) Hydroxy-isocromans (6) Lignans (Pinoresinol, Acetoxypinoresinol)	213–450	[5,6,7]
Tocopherols: α-tocopherol β-tocopherol γ-tocopherol	150–250 mg/kg 15–20% (of the total amount of tocopherols) 7–23% (of the total amount of tocopherols)	[6,13]
Colored pigments: Chlorophylls	2.41–38.7	[6] [12]

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
