# Peer review of "Extra Virgin Olive Oil and Metabolic Diseases"

_ijms, 2024, doi:10.3390/ijms25158117_

Round 1

Reviewer 1 Report

Comments and Suggestions for Authors

The authors described that EVOO presents a multitude of healthful properties and acts as a key parameter of clinical nutrition. The authors review the positive effects of EVOO, but their discussion is insufficient to warrant publication as a review. Adding answers to the questions below will help ensure the importance of this review.

1. The authors should describe and illustrate the mechanism by which EVOO exerts its positive effects.

1-1) Does EVOO exert its physiological effects via receptors?

1-2) Is EVOO taken up into cells and functions as a second messenger?

1-3) Does EVOO enhance or inhibit intracellular signaling mediated by some ligand-induced receptors?

1-4) Does EVOO incorporate into cell membranes and affect cell function?

1-5)Does EVOO bind to nutrients and affect their absorption from the intestinal tract or metabolism in the blood?

Reviewer 2 Report

Comments and Suggestions for Authors

This review article describes EVOO's beneficial health effects, including anti-diabetic, hypolipidemic, anti-hypertensive, and anti-obesity. The review is of great biological and medical relevance. It is well-organized and well-referenced, including key proposed mechanisms for EVOO-mediated protective vascular actions. 

Minor comments: Table 1 needs to be improved. Specifically, please add the quantity of each EVOO component at least in % i.e. fatty acid composition, minor components, sterols, polyphenols, and flavonoids, among others.

Comments on the Quality of English Language

Minor editing is required.
